# Pharmacological, Nutritional, and Rehabilitative Interventions to Improve the Complex Management of Osteoporosis in Patients with Chronic Obstructive Pulmonary Disease: A Narrative Review

**DOI:** 10.3390/jpm12101626

**Published:** 2022-10-01

**Authors:** Alessandro de Sire, Lorenzo Lippi, Vittorio Aprile, Dario Calafiore, Arianna Folli, Francesco D’Abrosca, Stefano Moalli, Marco Lucchi, Antonio Ammendolia, Marco Invernizzi

**Affiliations:** 1Physical and Rehabilitative Medicine Unit, Department of Medical and Surgical Sciences, University of Catanzaro “Magna Graecia”, Viale Europa, 88100 Catanzaro, Italy; 2Physical and Rehabilitative Medicine, Department of Health Sciences, University of Eastern Piedmont “A. Avogadro”, 28100 Novara, Italy; 3Dipartimento Attività Integrate Ricerca e Innovazione (DAIRI), Translational Medicine, Azienda Ospedaliera SS. Antonio e Biagio e Cesare Arrigo, 15121 Alessandria, Italy; 4Division of Thoracic Surgery, Department of Surgical, Medical and Molecular Pathology and Critical Care Medicine, University of Pisa, 56122 Pisa, Italy; 5Physical Medicine and Rehabilitation Unit, Department of Neurosciences, ASST Carlo Poma, 46100 Mantua, Italy

**Keywords:** osteoporosis, chronic obstructive pulmonary disease, bone health, sarcopenia, pulmonary rehabilitation, physical exercise, physical activity, dietary supplements, corticosteroids, rehabilitation

## Abstract

Osteoporosis is a highly prevalent condition affecting a growing number of patients affected by chronic obstructive pulmonary disease (COPD), with crucial implications for risk of fragility fractures, hospitalization, and mortality. Several risk factors have been identified to have a role in osteoporosis development in COPD patients, including corticosteroid therapy, systemic inflammation, smoke, physical activity levels, malnutrition, and sarcopenia. In this scenario, a personalized multitarget intervention focusing on the pathological mechanisms underpinning osteoporosis is mandatory to improve bone health in these frail patients. Specifically, physical exercise, nutritional approach, dietary supplements, and smoke cessation are the cornerstone of the lifestyle approach to osteoporosis in COPD patients, improving not only bone health but also physical performance and balance. On the other hand, pharmacological treatment should be considered for both the prevention and treatment of osteoporosis in patients at higher risk of fragility fractures. Despite these considerations, several barriers still affect the integration of a personalized approach to managing osteoporosis in COPD patients. However, digital innovation solutions and telemedicine might have a role in optimizing sustainable networking between hospital assistance and community settings to improve bone health and reduce sanitary costs of the long-term management of COPD patients with osteoporosis.

## 1. Introduction

Chronic obstructive pulmonary disease (COPD) is a burdensome pathological condition affecting approximately 10% of patients between 30 and 79 years old [1]. According to the World Health Organization, over 65 million people suffer from COPD worldwide with detrimental consequences in terms of global health and social costs [2]. Due to the negative impact of persistent respiratory symptoms on physical function and activity of daily living, a growing trend in disability was underlined by the Global Burden of Disease Study in 2019 in these patients [3]. In particular, COPD represents the 6th cause of increased global disability-adjusted life-years [3]. Moreover, given the functional consequences related to physical impairment and the high prevalence of comorbidities affecting these patients, it is not surprising that COPD might be frequently associated with both osteoporosis and physical frailty [2,4,5]. 

Osteoporosis is a critical age-related condition affecting a growing number of patients worldwide [6,7]. Specifically, twenty-two million European women and 5.5 million European men were estimated to be affected by osteoporosis [8], considered as a symptomatically silent condition until a fragility fracture occurs [9]. Fragility fractures are currently considered one of the most disabling symptoms of osteoporosis, resulting from low-energy trauma, such as a fall from a standing height or less [10]. Due to the aging of the population and the increase in osteoporosis prevalence, it has been estimated that fragility fractures will increase in the future years with detrimental consequences on both health outcomes and sanitary costs [9]. Therefore, effective preventive strategies are needed to reduce fracture risk in patients with osteoporosis [9,11]. 

In this scenario, a recent meta-analysis underlined that more than one-third of COPD patients (37.62%) might suffer from osteoporosis [12]. Moreover, it has been reported that COPD patients have a higher risk of osteoporosis compared to the general population due to the several risk factors involved in bone health impairment [13]. In particular, long-term corticosteroid therapy, systemic inflammation, smoke habitus, alcohol consumption, reduction in physical activity levels, malnutrition, and sarcopenia might have detrimental consequences on bone health and unfortunately have a high prevalence in COPD patients [13]. Moreover, glucocorticoid treatment, sarcopenia, low body weight, and smoking represent independent risk factors for risk of falls and fragility fractures [14,15]. 

Although a precise identification of the risk factors for fragility fractures should be the cornerstone of osteoporosis management [16], there is still a gap of knowledge in the current literature about the role of a comprehensive approach targeting the multilevel mechanisms underpinning osteoporosis in COPD patients [5]. 

Despite the fact that several modifiable risk factors have been identified to have a role in osteoporosis prevention in COPD patients [17], several barriers still affect the lifestyle approach to osteoporosis in this patient, while a personalized osteoporosis approach is still far from being fully characterized [18,19].

Indeed, to the best of our knowledge, no previous study summarized the approach to treat osteoporosis in COPD patients to provide evidence about sustainable strategies to improve bone health in the multidisciplinary management of COPD patients.

Thus, the aim of this narrative review was to provide a broad overview about the currently available pharmacological, nutritional, and rehabilitative approaches to treat osteoporosis in COPD patients in order to promote a personalized strategy for the multidisciplinary management of these frail patients.

## 2. COPD and Osteoporosis: A Close Link beyond Corticosteroids and Inflammation

To date, it is widely accepted that COPD and osteoporosis are closely linked. As shown in Figure 1, several risk factors might coexist and crucially affect osteoporosis risk in COPD patients, including long-term therapy, systemic inflammation, smoke, reduction in physical activity levels, malnutrition, and sarcopenia [17]. However, to date, the multilevel interactions between different risk factors have not been fully understood yet.

Long-term management of COPD patients frequently requires the use of inhaled CS (ICSs) or systemic CSs to improve pulmonary symptoms during both the maintenance phase and exacerbations [20]. To date, it is widely accepted that the detrimental consequences of CSs on bone health might be related not only to glucocorticoid long-term therapies, but even a single dose of glucocorticoids might increase fracture risk [21]. In addition, chronic systemic inflammation is a key component of COPD affecting not only molecular pathways involved in the etiology of the disease but also severity, functional outcomes, and complications risk. However, bone metabolism might be crucially affected by pro-inflammatory cytokines interacting with several pathways involved in bone remodeling [22]. 

Although the biological mechanisms underpinning osteoporosis development in COPD patients have not been fully characterized yet, other crucial factors should be considered in frail patients with COPD. 

Cigarette smoke is one of the most important risk factors for COPD [23]. On the other hand, smoking is an independent risk factor for osteoporosis and fragility fractures, with a cumulative effect over time [24]. Interestingly, it has been reported that nicotine might directly reduce both osteogenesis and angiogenesis, which play a key role in bone metabolism [25]. 

Concurrently, in COPD patients a progressive decrease in physical activity proportional to symptom progression has been reported, leading to a steeper decline in overall physical health status, promoting a vicious cycle that results in a progressive functional impairment [2]. On the other hand, physical activity plays a pivotal role in maintaining bone health given the role of mechanical stress stimuli on bone tropism and the positive effects of physical activity on skeletal muscle and bone–muscle crosstalk [24,26,27]. Moreover, it has been estimated that approximately 45% of COPD patients might have a compromised nutritional status related to the imbalance between energy expenditure and energy intake [28]. In this scenario, low body weight has been a widely recognized independent risk factor for low bone mineral density (BMD), fragility fractures, and sarcopenia [21,29]. 

In addition, sarcopenia is a complex and highly disabling condition that might affect about 15–55% of patients with COPD [30]. Interestingly, several risk factors have been identified to have a role in sarcopenia onset in people with COPD, including alcohol consumption, cigarette smoking, low physical activity levels, poor diet, and older age [29,31]. Moreover, impaired muscle quality has been frequently identified in COPD patients due to altered oxidative capacity and muscle mitochondrial function, with a shift toward glycolytic muscle fiber distribution, reduced capillary density, and reduced cross-sectional area [32]. 

On the other hand, it should be noted that growing literature recently highlighted a close link between muscle tissue and bone health [33]. Given the functional, embryological, and biochemical linkages between these two tissues, it is not surprising that several pathways involved in sarcopenia onset might be strictly related to osteoporosis in terms of osteosarcopenia [29,34,35,36]. 

In addition, sarcopenia might be also considered an independent risk factor for fragility fractures given its effects on physical performance characterizing sarcopenia severity [37], with crucial implications for balance and risk of falling [38]. The coexistence of sarcopenia and osteoporosis might crucially affect physical function and physical performance in COPD patients, already characterized by physical frailty and functional impairment [16,39]. In this contest, sarcopenia and osteoporosis might exponentially worsen disability, health-related quality of life (HR-QOL), and the need for assistance in COPD patients, with detrimental implications for social and sanitary costs [40]. 

Taken together, these findings highlighted that osteoporosis and COPD are two chronic disabling conditions that share several risk factors. However, there is still a large gap of knowledge about the role of single risk factors in osteoporosis development in patients with COPD. Moreover, most of the studies currently available assessed multiple associations of factors to have a role in osteoporosis development since the difficulties in isolating single factors [17,22,24,30]. On the other hand, a specific assessment of the most important contributors to osteoporosis development should be mandatory to set up a personalized multitarget intervention aiming at preventing osteoporosis and maintaining bone health. This concept could have relevant implications not only to reduce the disabling sequelae of osteoporosis but also to optimize the comprehensive management of both diseases.

## 3. Physical Activity in COPD

As previously reported, COPD patients show a significant reduction in physical activity levels compared to the general population, while common programs aiming at achieving 150 min of moderate-intensity physical exercise per week might be challenging in COPD patients [41].

As a result, a personalized approach should be considered to target the multicomponent disability affecting COPD patients (see Figure 2 for further details), in order to overcome the barriers to the implementation of effective and sustainable multidisciplinary therapeutic interventions triggering the multicomponent mechanisms underpinning osteoporosis development in COPD patients [42]. 

### 3.1. Pulmonary Rehabilitation

In 2013, the American Thoracic Society/European Respiratory Society defined PR as a “comprehensive intervention based on a thorough patient assessment followed by patient-tailored therapies, which include, but are not limited to, exercise training, education, and behavior change, designed to improve the physical and psychological condition of patients with chronic respiratory disease and to promote the long-term adherence of health-enhancing behaviors” [43]. Previous studies [44,45] showed that PR is one of the most effective interventions in reducing dyspnea, improving physical performance, and independence in activity of daily living in patients affected by COPD.

A pool of desirable components have been previously proposed to have a role in the comprehensive management of COPD patients. These include educational interventions, airway clearance techniques, inspiratory muscle training, and action plans for frequent exacerbations [46]. Interestingly, endurance training and strength training are key components of PR programs but have also a key role in the lifestyle approach to osteoporosis [47,48]. Moreover, a comprehensive PR approach might contribute to relieving dyspnea and reduce the risk of exacerbation, hospitalization and need for medications [49,50,51]. Thus, PR should be considered in a personalized approach providing additional synergisms between standard physical exercise, enhancing participation and patient engagement, and minimizing barriers that still affect physical activity in COPD patients. 

### 3.2. Rehabilitation and Therapeutic Exercise

Rehabilitation should be considered a key component of the comprehensive bone health treatment framework for COPD patients. In this scenario, the International Osteoporosis Foundation supports the effects of weight bearing, progressive resistance exercise, strength training, balance training, and tai-chi in both the prevention and treatment of osteoporosis [52]. On the other hand, a personalized exercise approach should be considered in COPD patients, which might present several barriers to standard programs commonly proposed for the general population [18]. 

In particular, it has been reported that endurance training is the most common exercise type used in COPD patients. However, continuous endurance training at the same intensity might exacerbate COPD patients symptoms such as dyspnea [53]. In this context, interval training might be the most suitable option given its reduced impact on pulmonary dynamics and the lower metabolic and ventilatory stress for COPD patients, which allows longer exercise time, and a longer capacity to sustain high-intensity exercise [45,54]. 

Exercise modalities might include biking or walking on a treadmill or floor, while the exercise intensity should be tailored to patient’s functional status and ranging between 4 and 6 on the modified Borg scale to safely achieve progression [45,54]. Exercise programs should be performed 3–4 times a week starting from 15 min per session and progressively increasing to reach 45–60 min per session of interval training [54,55]. Although endurance training has a key role in reconditioning skeletal muscle to oxidative metabolism [56], strength training is the most supported exercise modality in sarcopenic patients to prevent muscle loss and improve both muscle mass and strength [29,37,57]. Moreover, significant advantages of strength training were previously underlined in terms of exercise tolerance due to lower dyspnea occurrence during the training session [54]. Concurrently, exercise training should focus on hip and trunk muscles to stimulate osteogenic effects induced by mechanical stimuli and reduce both hip and vertebrae fracture risk, the most common sites of fragility fractures [58,59]. In addition, a recent meta-analysis emphasized the need for integrating balance control in the comprehensive rehabilitation treatment of COPD patients since a meaningful balance impairment is often reported and it might crucially increase the risk of falls and fragility fractures [60]. 

Specifically, over the past decades, growing literature underlined that people with COPD might have a balance impairment that cannot be explained by age-related processes alone, resulting in an increased risk of falls and fragility fractures [61,62]. In this scenario, several other risk factors have been identified to have a role in increased risk of falls in COPD patients, including skeletal muscle dysfunction, cognition, tremulousness, and vision [61]. However, given the high incidence of falls in COPD patients and the detrimental consequences on mobility, mortality, and assistance costs, a precise assessment of falls risk should be performed, aiming at developing effective physical exercises programs targeting the modifiable factors in patients with higher risk [61]. 

Furthermore, it is mandatory to consider all the potential approaches of rehabilitation (e.g., physical exercise, oxygen-ozone therapy, instrumental physical therapies, nutraceuticals, etc.) for the reduction in inflammation that could be confirmed by the lower serum levels of inflammatory cytokines [5,63,64,65,66]. In this context, it should also be considered the key role of the awareness about the benefits and the safety of physical exercise in the context of a personalized educational approach for patients with musculoskeletal disorders and COPD [5,63,64,65,66]. The benefits and the safety of individualized exercise intervention should be emphasized to promote the maximum compliance with the exercise programs [65]. Lastly, home-based exercise programs might be proposed, with recent reports underlining that telemedicine solutions might have a role in telemonitoring the therapeutic effects and personalizing the rehabilitation approach to patients needs [5]. Further details about digital innovation and telemedicine solutions have been characterized in the Section 7. 

### 3.3. Barriers and Challenges for the Physical Exercise Programs

Despite the positive effects of PR on the overall well-being and bone health of COPD patients, several barriers still affect PR spreading in routine clinical practice [5]. These barriers might be related to patients’ symptoms, psychological and behavioral factors, environmental barriers, and organizational models [67]. 

#### 3.3.1. Hypoxia and Oxygen Therapy

COPD patients commonly show lung hyperinflation and alveolar hypoxia, which might directly impact the exercise capacity, decreasing exercise tolerance and promoting inactivity. On the other hand, the cardiac output might be indirectly reduced in COPD patients due to the lower preload and increased resistance of pulmonary vasculature [67].

Therefore, the American Association of Cardiovascular and Pulmonary Rehabilitation suggested that oxygen saturation should be strictly monitored during physical activity and maintained at over 88%. In contrast, lower saturations require personalized prescriptions and oxygen supplementation during the exercise session. One’s own portable oxygen system might provide advantages in oxygen administration during physical activity, preventing desaturation and improving the feasibility of exercise programs. Concurrently, other pharmacological therapies might be considered to optimize exercise participation and safety. In this context, long-term adherence to physical exercise is a critical issue in literature, with several reports underlining the challenge of sustainable strategies for improving physical activity levels in COPD patients [68]. 

#### 3.3.2. Anemia 

Anemia might frequently characterize patients with chronic diseases, with detrimental consequences in COPD patients in terms of both fatigue and dyspnea [69]. Interestingly, a recent prospective study reported that anemia affects approximately 32.8% of COPD patients undergoing non-invasive ventilation, with a negative impact on physical function and HR-QOL [70]. On the other hand, it should be noted that in the current literature consensus about the precise red-cell hematic cut-off values is lacking and a recent systematic review failed to assess the exact prevalence of anemia in COPD patients [69]. Despite these considerations, clinicians should be aware about the negative effects of anemia in exercise participation and personalized therapeutic strategies should be performed to optimize the adherence to exercise programs in patients with COPD. 

#### 3.3.3. Adherence to Rehabilitation Programs 

Despite the widely documented positive effects of physical exercise in COPD patients, PR remains regrettably underutilized worldwide [71]. Beyond the physical and pathological barriers affecting COPD patients’ participation, each patient’s behavioral attitude greatly impacts on the compliance with rehabilitation programs [72]. Interestingly, the recent systematic review by Robinson et al. [73] suggested that HCPs have a key role in enhancing program participation, with positive effects of continued interaction and feedback with COPD patients. In addition, self-monitoring and participation in physical activity groups might further enhance patients’ adherence to the exercise programs. Therefore, a personalized approach should focus not only on the type of exercise prescription but also on patients’ individual compliance by providing suitable exercise options in different settings and social support in order to overcome barriers to COPD patient’s participation and improve the long-term management of these frail patients [73]. 

#### 3.3.4. Environmental and Organizational Issues

Environmental factors play a crucial role in PR availability and PR delivery in COPD patients, and a recent report underlined that great distances from patients’ homes and outpatient clinics, transportation problems and economic costs might represent barriers extremely difficult to overcome [5,71,72]. In addition, even though PR inpatients’ intensive services might guarantee effective PR programs in both acute and subacute patients, the outpatient referral and community settings might not be effective in the management of chronic COPD patients [71,74]. Beyond PR programs, osteoporosis management is historically underrecognized and underestimated in these patients despite the social and economic burden of fragility fractures [75]. In this scenario, it has been proposed that a close networking between hospital and community settings might improve the long-term management of frail patients with COPD [5]. On the other hand, specific educational programs increasing awareness in HCPs and physicians might increase patient engagement and participation in effective programs aiming at achieving comprehensive management of osteoporosis [5,71,72]. 

#### 3.3.5. Sustainability 

Sustainable strategies in the long-term management of COPD patients are a critical issue in current literature, with a growing number of reports highlighting the need for feasible models to improve respiratory function and quality of life of frail COPD patients [5,76,77]. Promising models of health care delivery have been proposed to address the clinical and organizational gap in the comprehensive rehabilitation management of COPD patients. In recent years, home-based rehabilitation and telerehabilitation interventions have been proposed to improve PR spreading and reduce sanitary costs, especially during the COVID-19 pandemic [72]. However, even these possibilities are not free from limitations, which include difficult technological approaches in older subjects, lack of standardization of healthcare delivery and patient assessment and health-professionals’ lack of training [78]. Even considering these limitations, digital innovation and telemedicine are promising approaches for personalized programs, monitoring and enhancing the effects of PR even in frail patients [5].

## 4. Nutraceuticals and Dietary Supplements

In recent years, growing attention has been raised on nutritional interventions tailored to COPD patients, with increasing evidence highlighting the synergistic role of these interventions with rehabilitation in enhancing the functional improvement induced by rehabilitation alone in frail patients [57,79]. In this scenario, a combined rehabilitation-nutrition therapeutic approach has been recently proposed to optimize lifestyle interventions targeting bone health in frail patients [79,80].

### 4.1. Vitamin D and Calcium Supplementation

Vitamin D is a fat-soluble secosteroid that plays a crucial role in calcium homeostasis, immune regulation and inflammatory response and is a cornerstone of osteoporosis treatment [81,82]. Interestingly, recent research has shown that COPD patients had an increased risk of vitamin D deficiency compared to the age-matched population [83]. The reasons underpinning the higher risk of vitamin D deficiency might be related to a lower food intake, aging, staying indoors, increased vitamin D catabolism due to corticosteroid therapies, impaired renal activation, and lower storage capacity in both muscles and fat tissues due to skeletal muscle wasting [83,84,85].

However, according to the National Osteoporosis Foundation, the International Osteoporosis Foundation, and the American Geriatric Society, a minimum of 30 ng/mL of vitamin D serum levels is needed to reduce the risk of falls and fragility fractures [86,87,88]. Moreover, the evidence supporting vitamin D supplementation in bone health management comes from large clinical trials highlighting significant benefits on bone mineral density, while controversies are still open about the effects of vitamin D supplementation on fragility fractures [89,90,91,92]. 

In the past few years, it has been shown that vitamin D might play a key role in skeletal muscle function and physical performance, which might frequently be impaired in COPD patients [93]. Therefore, vitamin D might have a key role in a comprehensive rehabilitation approach to COPD-related disability targeting physical performance and pulmonary function [94,95,96]. In addition, to the best of our knowledge, vitamin D and calcium supplementation has been associated with any pharmacological drug prescribed in COPD patients with osteoporosis and should be considered a milestone of the general therapeutic approach to osteoporosis [97,98,99,100,101,102,103].

Beyond the widely documented effect on bone reabsorption related to calcium absorption, excretion, and parathyroid hormone (PTH) regulation, vitamin D in COPD patients might target the multilevel biological pathways involved in osteoporosis development. On the other hand, recent research underlined that vitamin D deficiency might be related to cardiovascular [104], metabolic [105] and autoimmune disorders [106] and increased risk of cancer and cancer-related comorbidities [107]. Interestingly, during the COVID pandemic, increasing interest has been raised in vitamin D supplementation in patients with respiratory diseases due to the modulation of innate immune responses with relevant implications on the acute respiratory infection risk [108,109,110]. In this scenario, the recent meta-analysis by Jolliffe et al. [111] showed positive significant effects of Vitamin D (400–1000 IU vitamin D daily) supplementation in reducing the prevalence of acute respiratory infections. 

Taken together, these findings underline that vitamin D supplementation should be considered in the personalized management of patients with COPD, not only to improve bone health status but also to increase physical performance of the whole skeletal muscle system, which is commonly impaired in COPD patients. Moreover, due to the systemic effects of vitamin D on the immune response, further benefits might be experienced by patients with COPD, including reduced risk of acute respiratory infections and hospitalization. 

### 4.2. Proteins and Amino Acids

It has been estimated that approximately 45% of COPD patients might have compromised nutritional status mainly related to chronic systemic inflammation, oxidative stress and corticosteroid therapies characterizing the disease [112,113]. As a result, almost 50% of the patients with severe COPD might experience unintentional weight loss [28]. Due to the high prevalence of malnutrition and unintentional weight loss and its role as an independent risk factor for fragility fracture, a comprehensive approach including nutritional counseling and supplementation should be considered to improve the outcomes in frail COPD patients [28,112,113,114]. In this scenario, a previous meta-analysis supported the positive effects of nutritional supplementation on weight gain, body composition and physical performance [115]. On the other hand, it should be noted that inadequate protein intake might significantly affect bone health, negatively interacting with the calcium-phosphate metabolism [116]. Moreover, protein supplementation might improve insulin-like growth factor-I (IGF-I) serum levels, with positive significant effects on skeletal muscle anabolic stimulation [117].

Similarly, it has been proposed that aromatic amino acids might further promote growth factors regulating bone anabolisms, with potential implications on bone size, bone mass and bone strength [118,119]. In line with these considerations, sarcopenia is highly prevalent in COPD patients and should be targeted by a personalized nutritional approach including adequate dietary protein intake. In addition, a recent report underlined that dietary supplements might have a positive role in age-related conditions interacting with oxidative stress and mitochondrial dysfunction frequently characterizing COPD patients [120]. However, a recent systematic review underlined that there is insufficient evidence to support the integration of nutritional interventions with PR in COPD patients due to the heterogeneity of COPD clinical presentation and dietary supplements, outcome measures and PR programs [121].

### 4.3. Potential Role of Microbiota: The Gut–Bone Axis

In recent years, a growing number of reports suggested a strict link between gut microbiome and several chronic pathological conditions, such as the obesity and the osteoporosis [122,123]. The gut microbiome comprises approximately 10–100 trillion microorganisms living in mutually beneficial partnership with the human host, although recent research highlighted that gut microbiome alterations play a role in the pathogenesis and progression of several pathological conditions [124]. In COPD patients, smoking has detrimental effects on the gut microbiome, with recent reports highlighting that tobacco might lower the immunological system defenses, leading to increased infections and influencing the gut biofilm formation. Smoking might also negatively impact the gut environment, in particular oxygen tension and pH, modulating and selecting the proliferation of specific microorganisms [124]. In addition, COPD exacerbation and repeated antibiotic therapies might have detrimental effects on gut microbiota [125]. Concurrently, the gut microbiota might be strongly associated with diet and lifestyle habits, which are commonly impaired in patients affected by osteoporosis [122]. 

Despite these considerations, the gut microbiome might crucially affect bone health because of its role in regulating systemic inflammation [122,125]. In particular, gut microbiota function might contribute to the digestion of otherwise undigestible dietary fibers. Short-chain fatty acids derive from these processes, which are thought to have an anti-inflammatory action and modulate the immune systems [126]. Moreover, gut microflora dysregulation has been associated with changes in osteoclast activity, related to OPG/RANKL osteoclast pathway, intestinal microbiota regulation of serum IGF-1 and calcium intestinal absorption. Taken together, all these gut microbiome modifications might correlate with lower bone strength and quality [122,127]. It has been underlined that probiotics consumption might increase the calcium absorption and help to reshape the metabolic profile in obese subjects [122,123]. Moreover, prebiotics might increase colon concentrations of butyrate, mediating the positive effects of gut microbiota on bone metabolism [122,127]. Some examples of prebiotics used in clinical practice are galactooligosaccharides, fructooligosaccharides, inulin, digestion-resistant starch, xylooligosaccharides, and lactulose [122,127].

In conclusion, the evidence highlighted that “gut-bone axis” might be significantly impaired in COPD patients due to smoking behavior, systemic inflammation, and antibiotic therapies [122]. On the other hand, microbiome modulation might be part of the comprehensive management of bone health due to the multilevel interactions with different molecular pathways involved in bone remodeling. Moreover, interventions targeting the gut microbiome might show synergistic effects with other lifestyle interventions commonly used in bone health management, such as the interaction with macro- and micronutrient absorption and the mutual influence with physical exercise. 

## 5. Lifestyle Approach and Smoke Cessation 

Recent research highlighted that sanitary education towards a healthier lifestyle should be a cornerstone of PR, aiming at increasing both patients’ awareness about the negative effects of unhealthy behavior and the crucial role of the lifestyle medicine approach in improving their condition [128]. In COPD patients, smoke cessation appears to be the main intervention for arresting the progression of the disease, improving both the overall survival and reducing the risk of long-term complications [128]. On the other hand, smoke cessation plays a pivotal role also in bone health management given the detrimental effects of nicotine in terms of BMD loss [17]. In this scenario, a recent meta-analysis by Vestergaard et al. [129] highlighted that smoke cessation significantly reduces the risk of fractures in osteoporotic patients [129]. Moreover, a comprehensive rehabilitation approach might also include psychological support in order to overcome the psychological barriers to smoking cessation, and a recent Cochrane systematic review showed that group therapy is the most effective psychological support modality [130]. 

Beyond smoke cessation, patients should be instructed to reduce their sedentary behavior, especially elderly COPD patients, in light of the molecular anti-aging role of physical activity and the beneficial effects on BMD [131,132]. National and international recommendations recommend that physical exercise programs should be included in the comprehensive lifestyle approach to COPD to improve symptoms of these patients [131]. Furthermore, in sarcopenic older patients with fragility fractures, a comprehensive personalized approach, including nutritional and rehabilitative interventions, might be the most promising in terms of both symptoms and outcomes improvement [133]. 

Taken together, these findings emphasize the key role of a personalized lifestyle approach to bone health in COPD patients tailored to both patients’ characteristics and environmental factors. On the other hand, it should be noted that a lifestyle approach might be combined with pharmacological treatments in COPD patients with a higher risk of fracture to reduce the fragility fracture risk and their detrimental consequences on healthcare and sanitary costs.

## 6. Pharmacological Treatments

Pharmacologic treatment is a cornerstone in the therapeutic management of osteoporosis for prevention and treatment in patients at higher risk of fragility fracture [134]. However, due to the lack of specific evidence in COPD patients, the main recommendation follows general practice guidelines for the treatment of primary osteoporosis [21]. In addition, most of the evidence about the pharmacological management of osteoporosis mainly focuses on women, while COPD disease is highly prevalent in men due to the strict link with smoking behavior. On other hand, different studies have investigated the efficacy of anti-resorptive drugs and anabolic drugs in the treatment of glucocorticoid-induced osteoporosis (GIO) [135,136,137]. Approved drugs for the treatment of osteoporosis include anti-resorptive drugs such as bisphosphonates, denosumab, and the anabolic agent teriparatide [138]. Specifically, anti-resorptive drugs primarily inhibit osteoclastic bone resorption with later secondary effects on bone formation, while anabolic drugs stimulate osteoblastic bone formation with variable effects on bone resorption [138,139]. 

Unfortunately, to the best of our knowledge, no previous study compared the effectiveness of different pharmacological therapies in patients with COPD, and the optimal pharmacological strategy showing more clinical efficacy in COPD patients is still far from being fully characterized.

However, Table 1 summarized the potential treatments proposed to have a role in the comprehensive osteoporosis management of COPD patients, including pharmacological treatments.

Bisphosphonates (BPs) are the most prescribed drugs in the management of primary and secondary osteoporosis, including GIO [140,141]. BPs are analogs of inorganic pyrophosphate and inhibit bone resorption. They can block osteoclastic activity, but they cannot reinstate lost structure or improve bone micro-architecture, given their inability to stimulate osteoblast activity [134,139,142]. Therefore, oral BPs, such as alendronate and risedronate, are effective as first-line therapies in preventing fractures in GIO, combined with an adequate calcium and vitamin D supplementation [143]. Alendronate 70 mg once weekly or 10 mg daily is recommended for the treatment of women with postmenopausal osteoporosis, men with osteoporosis, and GIO, reducing the incidence of vertebral, non-vertebral, and hip fractures [144]. In COPD patients, the RCT by Smith et al. [145] investigated the effect of alendronate compared to placebo combined with calcium supplements of 600 mg/d, for 12 months in both groups, reporting a significant improvement in lumbar BMD in the alendronate group. 

Similarly, Risedronate 35 mg once weekly or 5 mg daily has been shown to reduce vertebral and non-vertebral fractures, and it is approved for the treatment of osteoporosis [98,99,100].

In contrast, Ibandronate 150 mg once monthly is recommended for the treatment of postmenopausal women with osteoporosis, reducing vertebral fracture incidence [146]. On other hand, to the best of our knowledge, no data are available to show the efficacy of hip fracture risk reduction [147,148]. Because of concerns over possible adverse effects of long-term bisphosphonate therapy (i.e., osteonecrosis of the jaw and atypical fractures), the need to continue treatment should be reviewed at regular intervals and the therapeutic treatment should be personalized on patients individual risk [149]. Indeed, it is crucial that an adequate screening of oral health status through a dentistry evaluation in high-risk subjects is carried out before administering anti-osteoporosis drugs [150,151]. In addition, these drugs might be commonly associated with adverse effects, including gastrointestinal problems, bowel disturbance, headache and musculoskeletal pain [152]. Therefore, a comprehensive assessment is needed in COPD patients, including early detection of adverse events including gastroesophageal symptoms that might be highly prevalent in COPD and frequently associated with acute exacerbations [153]. Based on the available data, it is recommended that the risk should be reassessed after 5 years for alendronate, risedronate, or ibandronate. Withdrawal of treatment from alendronate, risedronate, or ibandronate is associated with decreases in BMD and increased bone turnover after 2–3 years for alendronate and 1–2 years for ibandronate and risedronate [154]. 

In addition, to the best of our knowledge, there have been no head-to-head RCTs to show differences in anti-fracture efficacy between oral BPs and direct comparisons between individual RCTs, which are difficult because of heterogeneity in patient populations, study design and confounders [155].

Concerning the intravenous bisphosphonate, the Zoledronate (5 mg/i.v./year) was registered for treatment of GIO based on a study that clearly documented its positive effect on vertebral, nonvertebral, and hip fracture risk after 3 years of treatment [156,157]. As oral bisphosphonate, zoledronate is the most cost-effective first-line drug option for bone protection. A study of an extension of the treatment to 9 years showed that the bone mass values at the femoral level remained stable [158]. In frail patients, zoledronic acid might reduce the risk of new clinical fractures and mortality when administered 2 weeks after a hip fracture [159]. Zoledronate has also been approved in the management of osteoporosis in men taking glucocorticoids, based on bridging studies [157,160]. 

Side effects of zoledronic acid include an acute phase reaction, usually only after the first infusion, and gastrointestinal symptoms. Glomerular filtration rate (eGFR) should be calculated prior to initiation of treatment and caution should be advised for recipients at risk of kidney failure; monitoring for any increase in serum creatinine or reduction in eGFR. Symptomatic atrial fibrillation might be a serious adverse event reported in phase III validation trial [158]. 

### 6.1. Denosumab

Denosumab, a fully human IgG2 monoclonal antibody against Receptor Activator of Nuclear factor Kappa B (RANKL-B), has been proposed to treat osteoporosis patients by reducing the rate of clinical fragility fractures (both hip and vertebrae) [97]. Denosumab is administrated as a subcutaneous injection of 60 mg once every 6 months. It is approved for the treatment of postmenopausal women and men at increased fracture risk, and for the prevention and treatment of cancer treatment-induced bone loss [97,139]. In patients with GIO, denosumab has been associated with reduction in vertebral, non-vertebral, and hip fractures, while safety and efficacy are maintained over 10 years of treatment, also considering the high compliance and persistence by the patients [161,162,163]. 

Hypocalcemia is a common side-effect, increasing in patients with renal failure, and patients should be aware to report symptoms of hypocalcemia. In patients with a high risk of hypocalcemia (e.g., patients with a creatinine clearance <35 mL/min), serum calcium levels should be checked within 2 weeks after the initial dose [164]. Other side effects include skin infection, predominantly cellulitis, and eczema. Rare adverse effects of denosumab include osteonecrosis of the jaw and atypical femoral fractures [165]. 

Denosumab cessation leads to rapid reductions in BMD through the elevations in bone turnover [166]. In a post hoc analysis of the FREEDOM study, women discontinuing denosumab had an increased rate of vertebral fracture over an average of 3–6 months since the last denosumab injection. The increase in vertebral fracture risk following cessation of denosumab therapy emphasizes the need to consider continued treatment with an alternative anti-resorptive drug following denosumab withdrawal. 

### 6.2. Teriparatide

Teriparatide is a synthetic analog of PTH with anabolic skeletal effects, approved for the treatment of osteoporosis in postmenopausal women, in men with a higher risk of fragility fracture, and in patients with GIO (20 mg/day for 24 months) [101,102,103]. Interestingly, comparative studies showed a greater increase in lumbar and femoral neck BMD in patients receiving teriparatide compared with bisphosphonates [167,168]. In addition, teriparatide is effective in reducing both vertebral and non-vertebral fractures in postmenopausal women with osteoporosis [167]. Specifically, systematic reviews with meta-analysis have shown an OR for hip fracture risk of 0.44 (95% CI: 0.22, 0.87; *p* = 0.019) in patients treated with teriparatide compared with placebo [169,170]. 

Interestingly, teriparatide might positively affect even bone quality due to its anabolic effects on skeletal tissues. A recent RCT assessed the effects of teriparatide in terms of trabecular bone score, reporting a significant increase after 36 months in patients treated with teriparatide, but no significant changes were found in patients treated with alendronate [171]. Adverse drug effects include hypercalcemia, hyperparathyroidism, renal impairment, headache, nausea, dizziness, postural hypotension, leg pain, and teratogenic consequences. 

## 7. Sustainable Strategies and Digital Innovation 

Despite the growing evidence emphasizing the need for effective integration of personalized bone health management in COPD patients, a comprehensive approach to osteoporosis is still regrettably underestimated and underrecognized in COPD patients [172,173]. It has been proposed that it might be partly related to the lack of organizational models including specialized physicians managing osteoporosis in COPD patients, while the multidisciplinary team involved in COPD management and pulmonologist specialists generally focus on the primary symptoms related to the disease [19]. On the other hand, effective strategies are needed to address bone health in these patients and recent research has proposed new sustainable strategies potentially promoting the integration of bone health assessment in routine clinical practice. In particular, the recent study by Singhvi and Bon [174] underlined that chest CT scans might be useful to assess bone attenuation of the thoracic and first lumbar vertebrae, with a significant correlation to bone mineral density assessed with DXA measured. In addition, the authors suggested that low vertebral bone attenuation is associated with lung function and increased exacerbation frequency, highlighting that a precise bone quantity assessment might be part of a comprehensive assessment of COPD patients targeting not only bone health but also reflecting the whole physical wellbeing [174].

In addition, the COVID-19 pandemic has highlighted new challenges in the management of patients with COPD that were isolated for months at home with decreased physical activity levels and reduced health care delivery [175,176]. In this scenario, “telerehabilitation” has been proposed to improve the remote management of these frail patients; telemedicine spreads with the increasing interest of physicians on the technological innovation in rehabilitation during the COVID-19 pandemic [177,178,179,180]. 

Interestingly, several models have been proposed to integrate technological advances in COPD management, with promising results in terms of safety and validity. Specifically, the recent review by Stickland et al. [181] reported no significant differences between telerehabilitation and standard PR with intriguing implications in overcoming barriers to rehabilitation delivery in community settings [181]. In addition, telehealth might be a suitable option to reduce the sanitary costs of long-term COPD management, saving time spent by HCPs in visits or treatments, and concurrently reducing costs and time spent by patients to reach health institutions [182]. On the other hand, it should be noted that, in recent years, growing literature emphasized that standard PR should be considered a cost-effective therapy, with strong evidence supporting its positive effects in reducing health care resources and preventing exacerbation and hospitalization related to pulmonary complications in COPD patients [183]. Besides these considerations, PR still remains underestimated and underused with recent reports underlining that only 2.7% of US Medicare patients receive PR after 12 months from COPD exacerbation [184]. In this scenario, technological advances might boost telemedicine delivery in community settings, integrating the key component of PR in lifestyle approach of COPD patients. Specifically, a strict telemonitoring of COPD patient parameters might promote an early therapeutic intervention crucially affecting the risk of exacerbations. Moreover, digital innovation might improve exercise program delivery, might improve education of both patients and caregivers, and might support behavior changes with potential benefits in physical function and HR-QOL [46,185,186,187]. In this context, the RCT by Rutkowski et al. [188] introduced a virtual reality approach to improve PR delivery in patients with chronic pulmonary diseases, reporting several advantages of this approach.

Despite the fact that technological advances might overcome barriers to health care delivery, several limitations might affect patients’ engagement in telemonitoring, telemedicine and telerehabilitation programs [189,190,191]. Specifically, lack of access to technology or scarce technology skills might be a crucial limitation, especially in older adults. Similarly, patient characteristics and comorbidity including impairment in hearing, vision, communication, or cognitive functions might be critical issues impossible to overcome [192]. Moreover, to date, no previous study assessed the role of telemedicine in the osteoporosis management of COPD patients. However, the recent study by Gani et al. [193] underlined that telemedicine might improve the compliance rate to osteoporosis pharmacological therapies and prevent hip fractures in the elderly. However, growing evidence is supporting the positive effects of telemedicine in lifestyle approach, a widely accepted key component of the therapeutic management of osteoporosis [194]. 

Altogether, our findings suggested that telemedicine might be considered in the comprehensive rehabilitation approach to COPD patients with potential benefits on bone health management. However, specific organizational models are needed to promote effective networking between hospital assistance and community settings to improve bone health and reduce sanitary costs of the long-term management of these frail patients. 

## 8. Conclusions

Despite remarkable knowledge about the mechanisms underpinning osteoporosis development in COPD patients being acquired, the bone health management of these patients remains a challenge for physicians.

This narrative review highlighted that a multidisciplinary approach is mandatory to improve bone health and reduce the risk of fractures in COPD patients. Several different interventions should be integrated to optimize the bone health management in these patients, and an effective networking between hospital assistance and community settings should be improved, including a personalized multidisciplinary treatment for osteoporosis. Further studies are still needed to better characterize the effects of integrated and personalized treatments to improve the long-term management of osteoporosis in COPD patients.

## Figures and Tables

**Figure 1 jpm-12-01626-f001:**
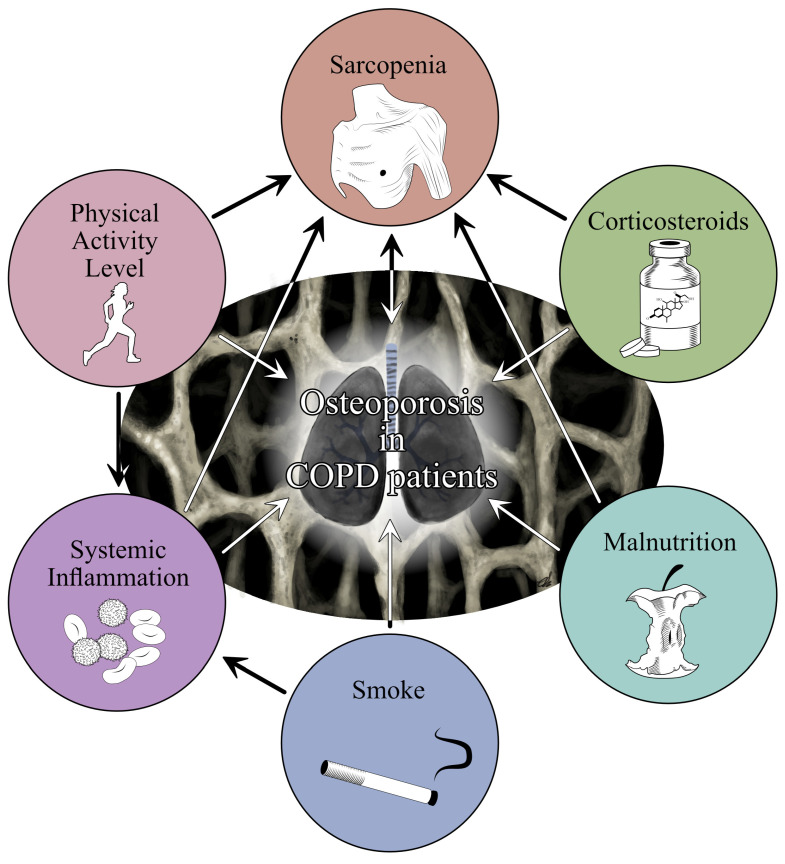
The figure summarized the multilevel interaction between the most common risk factors promoting Osteoporosis in COPD patients.

**Figure 2 jpm-12-01626-f002:**
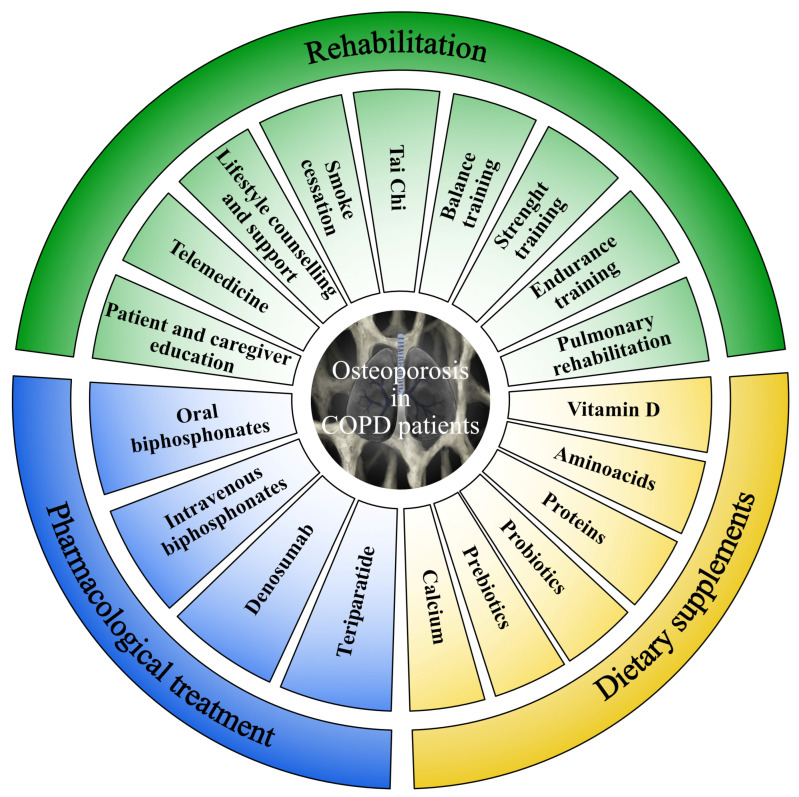
Personalized approach to target the multicomponent mechanisms underpinning osteoporosis development in COPD patients.

**Table 1 jpm-12-01626-t001:** Treatment possibilities in COPD patients with osteoporosis.

Area	Interventions	Goals
*Rehabilitation*		
Pulmonary rehabilitation	Educational interventions, airway clearance techniques, inspiratory muscle training, action plans for frequent exacerbations, endurance training, and strength training	Relieving dyspnea, reduce risk of exacerbation and hospitalization, and reduce need for medications
Exercise program	Weight bearing, progressive resistance exercise, strength training, balance training, tai-chi, endurance training, interval training	Reconditioning skeletal muscle, preventing muscle loss, and improving both muscle mass and strength
Lifestyle interventions	Patient and caregiver education, lifestyle counseling and support, smoke cessation	Increasing awareness about unhealthy behavior and improving symptoms through lifestyle changes
Sustainable programs	Telemedicine, home-based rehabilitation, telerehabilitation	Increasing adherence, improving PR spreading, and reducing sanitary costs
*Dietary supplements*		
Vitamin D	Oral Supplementation	Increasing bone health, skeletal muscle function and physical performance, positive stimulation of immune response
Calcium	Oral Supplementation	Increasing bone health, skeletal muscle function, and physical performance
Proteins	Oral Supplementation	Promoting weight gain, improving body composition, and physical performance. Increasing bone health and skeletal muscle mass
Aminoacids	Oral Supplementation	Promoting weight gain, improving body composition, and physical performance. Increasing bone health and skeletal muscle mass
Probiotics	Oral Supplementation	Promoting anti-inflammatory action and modulating immune systems. Promoting bone health
Prebiotics	Oral Supplementation	Promoting anti-inflammatory action and modulating immune systems. Promoting bone health
*Pharmacological treatment*		
Oral bisphosphonate	Alendronate (70 mg/week or 10 mg/day) Risedronate (35 mg/week or 5 mg/day) Ibandronate (150 mg/month)	Inhibiting bone resorption, reducing fracture risk
Intravenous bisphosphonate	Zoledronate (5 mg/year)	Inhibiting bone resorption, reducing fracture risk
Denosumab	Subcutaneous injection (60 mg/6 months)	Inhibiting bone resorption, reducing fracture risk
Teriparatide	Subcutaneous injection (20 mg/day for 24 months)	Promoting skeletal anabolism (with effect on BMD and bone quality), reduce fracture risk

*Abbreviations*: BMD: Bone Mineral Density, COPD: Chronic Obstructive Pulmonary Disease, PR: Pulmonary Rehabilitation.

## Data Availability

Not applicable.

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
