# Peer review of "Pharmacological, Nutritional, and Rehabilitative Interventions to Improve the Complex Management of Osteoporosis in Patients with Chronic Obstructive Pulmonary Disease: A Narrative Review"

_jpm, 2022, doi:10.3390/jpm12101626_

Round 1

Reviewer 1 Report

De Sire et al aimed to provide a broad overview of the currently available pharmacological, nutritional and rehabilitative approaches to treat osteoporosis in COPD patients in order to promote a personalized strategy for the multidisciplinary management of these frail patients using a narrative review approach. Management of osteoporosis in COPD is an incredibly important topic. However, I was disappointed overall with the presentation of this manuscript. This largely stemmed from the diffuse introduction which does not provide the rationale for treating osteoporosis in COPD differently than osteoporosis in general. Much of the introduction summarizes associated risk factors common to both COPD and osteoporosis without delving into why the common risk factors reflect common underlying etiology. Some of this information is provided later in the manuscript however, needs to be emphasized earlier to grab reader interest e.g. what smoking does to bones and lungs, osteoporosis prevalence is higher in COPD (particularly men) than the general population. The article also would benefit from extensive technical writing review to increase readability.   

Major:  

1)     What is a fragility fracture? How does this differ from a fracture in general? Why is it important to distinguish?

2)     “In particular, COPD represents the 6th cause of increased global disability-adjusted life-years [3], thus, it is not surprising that COPD has been frequently associated with physical frailty and osteoporosis.” Needs more elaboration. Frailty and osteoporosis are not synonyms for disability. Link between frailty and osteoporosis with disability needs to be more clearly spelled out.

3)     Introduction summarizes series of risk factor associations without digging into why the associations are indicative of common etiology.

4)     “the aim of this narrative review was to provide a broad overview about the currently available pharmacological, nutritional and rehabilitative approaches to treat osteoporosis in COPD patients in order to promote a personalized strategy for the multidisciplinary management of these frail patients.” Goal is disconnected from the introductory paragraphs which do not mention rationale for pharmacological, nutritional and rehabilitation approaches to osteoporosis in COPD. What is special about osteoporosis in COPD that is not captured by osteoporosis treatment in general? Why is there a need specifically in COPD? Is the prevalence of osteoporosis higher in COPD than non-COPD? Age of onset of osteoporosis sooner in COPD?

5)     Figure 2 is a very complicated approach. I am surprised little of the manuscript describes the disconnect between pulmonary and geriatric clinicians in terms of case management. Treatments exist for osteoporosis; however, COPD patients are largely seen by pulmonologists who do not treat osteoporosis or even screen for it.

6)     Patients with COPD get chest CTs which can be used to estimate bone density. This is an obvious tool that could be used for screening in the pulmonary setting. However, it’s not established as part of care management. This should be mentioned.

 Minor:

1)     Writing is awkward throughout with words missing in sentences and other issues. Needs technical writer to edit it.

2)     Define HRQL, PA

Author Response

Dear Reviewer

thank you for your letter and kind comments concerning our manuscript entitled “Pharmacological, nutritional and rehabilitative interventions to improve the complex management of osteoporosis in patients with chronic obstructive pulmonary disease”. We would like to express our sincere appreciation for your careful reviewing and invaluable comments which help us to further improve this paper.

Revisions based on your comments are highlighted in the manuscript in yellow, and our detailed responses according to each revision are shown as followed

Major:

1) What is a fragility fracture? How does this differ from a fracture in general? Why is it important to distinguish?

We would like to thank the reviewer for the insightful comment. We improved the Introduction Section by better clarifying the fragility fracture definition and differences between high energy and low energy fractures. Moreover, we better emphasized the importance of fragility fractures prevention to reduce negative consequences in health outcomes in accordance with the Reviewer’s instructions.

2) “In particular, COPD represents the 6th cause of increased global disability-adjusted life-years [3], thus, it is not surprising that COPD has been frequently associated with physical frailty and osteoporosis.” Needs more elaboration. Frailty and osteoporosis are not synonyms for disability. Link between frailty and osteoporosis with disability needs to be more clearly spelled out.

We would like to thank the reviewer for the insightful comment. We better characterized the link between disability, osteoporosis and frailty. Moreover, we improved the sentence in accordance with the Reviewer’s instructions.

3) Introduction summarizes series of risk factor associations without digging into why the associations are indicative of common etiology.

We would like to thank the reviewer for the insightful comment. We better underlined in the Introduction Section why the associations between risk factors and osteoporosis are indicative of common etiology. However, in order to avoid repetition in the manuscript, we shortly summarized this issue given it have been more detailed presented in the other sections. We hope that the Reviewer might understand the aim to be more concise in the Introduction section. However, if the Reviewer still prefers that we deal with this large topic in this section we will be glad to meet the reviewer's instructions.

4) “the aim of this narrative review was to provide a broad overview about the currently available pharmacological, nutritional and rehabilitative approaches to treat osteoporosis in COPD patients in order to promote a personalized strategy for the multidisciplinary management of these frail patients.” Goal is disconnected from the introductory paragraphs which do not mention rationale for pharmacological, nutritional and rehabilitation approaches to osteoporosis in COPD. What is special about osteoporosis in COPD that is not captured by osteoporosis treatment in general? Why is there a need specifically in COPD? Is the prevalence of osteoporosis higher in COPD than non-COPD? Age of onset of osteoporosis sooner in COPD?

We would like to thank the reviewer for the insightful comment. We better clarified that COPD patients have a higher risk for osteoporosis compared to general population due to the several risk factors presented. Despite several modifiable risk factors have been identified in COPD patients, several barriers still affect the lifestyle approach to osteoporosis in this patient, while a personalized approach is needed in COPD patients.

We would like to thank the reviewer for the opportunity to further improve the Introduction Section by clarifying these critical issues.

5) Figure 2 is a very complicated approach. I am surprised little of the manuscript describes the disconnect between pulmonary and geriatric clinicians in terms of case management. Treatments exist for osteoporosis; however, COPD patients are largely seen by pulmonologists who do not treat osteoporosis or even screen for it.

We would like to thank the reviewer for the insightful comment. We totally agree with this comment, the Reviewer underlined the unmet clinical need that brought us to realize this review. Moreover, it is surprising that still little effort has been paid in the scientific literature on this topic and osteoporosis assessment and treatment are still underrecognized and underestimated. In line with the Reviewer’s suggestion, we improved the “Sustainable strategies and digital innovation” Section better underlining this critical issue.

6) Patients with COPD get chest CTs which can be used to estimate bone density. This is an obvious tool that could be used for screening in the pulmonary setting. However, it’s not established as part of care management. This should be mentioned.

We would like to thank the reviewer for the insightful comment. This is a very interesting topic that is progressively gaining interest in literature to reduce the number of exams required in COPD patients and sanitary costs. Despite few studies focused on this approach in these patients, it is a suitable option to improve care management of people with COPD and it has been included in the Section “Sustainable strategies and digital innovation”.

We would like to express our sincere appreciation for this comment which help us to further improve this paper

Minor:

1) Writing is awkward throughout with words missing in sentences and other issues. Needs technical writer to edit it.

We would like to thank the reviewer for the insightful comment. We ask a technical writer to edit the text and the whole manuscript has been improved following the Reviewer’s instructions.

2) Define HRQL, PA

We would like to thank the reviewer for the insightful comment. As suggested, we adequately clarified all the abbreviations at their first appearance in the text.

Reviewer 2 Report

This is a general narrative review about a vague and controversial topic.

Literature search is adequate.

Probably I try to include more data about vitamin D and also about fractures in these patients.

Besides, I would try to quantity each factor influencing bone metabolism in the multifactorial development of bone damage in these patients.

Other topics would be to highlight what therapeutic agent showed more clinical efficacy in COPD patients.

Finally the topic of sarcopenia might be developed more comprehensively.

I hope these observations help to improve this interesting review.

Besides authors should include a more detailed references about falls in COPD patients.

Author Response

Dear Reviewer

thank you for your letter and kind comments concerning our manuscript entitled “Pharmacological, nutritional and rehabilitative interventions to improve the complex management of osteoporosis in patients with chronic obstructive pulmonary disease”. We would like to express our sincere appreciation for your careful reviewing and invaluable comments which help us to further improve this paper.

Revisions based on your comments are highlighted in the manuscript in yellow, and our detailed responses according to each revision are shown as followed.

Probably I try to include more data about vitamin D and also about fractures in these patients.

We would like to thank the reviewer for the insightful comment. We improved the introductions section by providing more data about fragility fractures in these patients. Moreover, we improved the “Vitamin D and calcium supplementation” Subsection better characterizing the higher incidence of Vitamin D deficiency in people with COPD and the need for adequate supplementation to counteract the systemic effects of vitamin D deficiency.

Besides, I would try to quantity each factor influencing bone metabolism in the multifactorial development of bone damage in these patients.

We would like to thank the reviewer for the insightful comment. The Reviewer highlighted a very interesting and controversial topic. Unfortunately, to the best of our knowledge, no prospective studies have quantified the role of single risk factors in osteoporosis development since the difficulties in isolating these factors. However, we totally agree that a specific assessment of the most important contributors to osteoporosis development is crucial in a personalized approach to osteoporosis in COPD patients. We significantly improved the manuscript in accordance with the Reviewer’s comment.

Other topics would be to highlight what therapeutic agent showed more clinical efficacy in COPD patients.

We would like to thank the reviewer for the insightful comment. Unfortunately, there is a lack of specific evidence about the optimal pharmacological approach for COPD patients. Moreover, to date, therapeutic agents showing more clinical efficacy in COPD patients are still far from being fully defined. Therefore, we better emphasize the gap of knowledge the Reviewer has suggested to further improve our manuscript.

Finally the topic of sarcopenia might be developed more comprehensively.

We would like to thank the reviewer for the insightful comment. We improved the sarcopenia characterization highlighting the need for a precise sarcopenia assessment in COPD patients in accordance with the Reviewer’s suggestion.

Besides authors should include a more detailed references about falls in COPD patients.

We would like to thank the reviewer for the insightful comment. We improved the “Physical activity in COPD” Section providing more detail about falls in COPD patients and balance impairment characterizing the disease, highlighting the role of rehabilitation in reducing  risk of falling.

Round 2

Reviewer 1 Report

The authors have addressed my concerns. Thank you. 

Author Response

Thank you for your time.